# CYP3A-Mediated Carbon–Carbon Bond Cleavages in Drug Metabolism

**DOI:** 10.3390/biom14091125

**Published:** 2024-09-05

**Authors:** Junhui Zhou, Xuan Qin, Shenzhi Zhou, Kevin R. MacKenzie, Feng Li

**Affiliations:** 1Center for Drug Discovery, Department of Pathology and Immunology, Baylor College of Medicine, Houston, TX 77030, USA; junhui.zhou@bcm.edu (J.Z.); xuan.qin@bcm.edu (X.Q.); shenzhi.zhou@bcm.edu (S.Z.); kevin.mackenzie@bcm.edu (K.R.M.); 2NMR and Drug Metabolism Core, Advanced Technology Cores, Baylor College of Medicine, Houston, TX 77030, USA; 3Department of Biochemistry and Molecular Pharmacology, Baylor College of Medicine, Houston, TX 77030, USA

**Keywords:** CYP3A, carbon–carbon bond cleavage, drug metabolism

## Abstract

Cytochrome P450 enzymes (P450s) play a critical role in drug metabolism, with the CYP3A subfamily being responsible for the biotransformation of over 50% of marked drugs. While CYP3A enzymes are known for their extensive catalytic versatility, one intriguing and less understood function is the ability to mediate carbon–carbon (C–C) bond cleavage. These uncommon reactions can lead to unusual metabolites and potentially influence drug safety and efficacy. This review focuses on examining examples of C–C bond cleavage catalyzed by CYP3A, exploring the mechanisms, physiological significance, and implications for drug metabolism. Additionally, examples of CYP3A-mediated ring expansion via C–C bond cleavages are included in this review. This work will enhance our understanding of CYP3A-catalyzed C–C bond cleavages and their mechanisms by carefully examining and analyzing these case studies. It may also guide future research in drug metabolism and drug design, improving drug safety and efficacy in clinical practice.

## 1. Introduction

Cytochrome P450 enzymes (P450s) represent a diverse and essential class of thiolate-ligated heme proteins characterized by their remarkable catalytic versatility [1,2]. These enzymes play pivotal roles in various physiological functions, spanning specialized biosynthetic pathways to xenobiotic detoxification [3]. Their catalytic repertoire includes a broad spectrum of oxidations crucial for cellular function, such as hydroxylation, dealkylation, deamination, dehalogenation, cyclization [4,5], carbon–nitrogen (C–N) bond formation [6], and C–C bond cleavage [7,8,9]. Understanding the mechanisms and substrate specificity of P450 enzymes is crucial for elucidating their roles in physiology and pharmacology, as well as for the development of novel therapeutic interventions and biotechnological applications [3]. P450s play a crucial role in the phase I metabolism of xenobiotics, accounting for approximately 75% of drug metabolism [10]. However, mammalian drug-metabolizing P450-mediated C–C bond cleavages are uncommon in drug metabolism, although P450-mediated C–C bond cleavages in sterol metabolism, including CYP51 [11,12], CYP11A [13], CYP17A [14], CYP19A [15], and CYP24A1 [16] and bacterial P450s (CYP125, CYP152L1, CYP152A1, and CYP107H1) catalyzed C–C bond cleavage in fatty acid metabolism, have been reported [17,18,19,20,21]. The exact mechanisms underlying these cleavages remain incompletely understood.

The CYP3A subfamily including CYP3A4 and CYP3A5 is a prominent member of the P450 family, mainly expressed in the liver and intestine [22,23]. CYP3A enzymes play a crucial role in the metabolism of a wide range of xenobiotics and endogenous compounds, contributing to the biotransformation of over 50% of marketed drugs [24,25,26,27]. The primary role of CYP3A in drug metabolism underscores its clinical significance, as variations in CYP3A activity can result in drug–drug interactions, influencing drug efficacy and toxicity [28]. CYP3A-involved C–C bond cleavages have been reported. Determining CYP3A-mediated unusual C–C bond cleavage reactions may improve drug efficacy and safety as well as elucidate the pathophysiology of various diseases [8,29]. This review focuses on CYP3A-mediated C–C bond cleavages with the proposed possible mechanisms in the previous report, which will provide insights for the scientists working on the P450-mediated uncommon reactions and drug development. The drugs included in this review are listed in Table 1.

## 2. CYP3A Mediates C(sp^3^)-C(sp^3^) Bond Cleavages

### 2.1. Plitidepsin

Plitidepsin (also known as Aplidin), a marine-derived cyclic depsipeptide, has shown potent antitumor activity against a variety of cancer cell lines, including melanoma, ovarian, renal, prostate, and breast cancers [31,33]. Its effectiveness in treating non-small-cell lung and colon cancers has made it a promising candidate in phase II clinical trials [32].

Previous metabolism studies of Plitidepsin in human liver microsomes (HLM) and human P450 isoenzymes indicated that CYP3A4-mediated C-dealkylation occurred leading to M1 (Figure 1) [30]. The exact mechanism of this C–C bond cleavage is not fully understood [30]. C-dealkylation metabolite was also observed in the urine and feces of patients treated with Plitidepsin [56]. More studies are needed to understand the cleavage. 

### 2.2. KRO-105714

KRO-105714 was developed due to its anti-atopic dermatitis (eczema) activity by robust suppression of the sphingosylphosphorylcholine (SPC) receptor [35]. Experiments with a specific CYP3A4 inhibitor (ketoconazole at 2 µM) and in HLM and human P450 isoforms revealed that CYP3A4 was the primary enzyme contributing to the formation of monohydroxylated (M2), O-demethylated, and C-demethylated KRO-105714 (M3) metabolites [34].

The metabolite (M3) formed by uncommon C–C bond cleavage was confirmed by its synthetic standard. In CYP3A4 isoform, alcohol M2 can be readily converted to M3. Thus, the pathway leading to C-demethylation likely involves the initial hydroxylation of KRO-105714 to generate an alcohol M2. The M2 was further oxidized to acid, which undergoes the elimination of CO_2_ to form M3 (Figure 2). In the biological system, other enzymes like decarboxylase may be involved in the M3 formation. The exact mechanism of the C–C bond cleavage is still not clear.

### 2.3. Noscapine

Noscapine is a non-sedating alkaloid used as an antitussive (cough suppressant) and for various cancers [57]. Due to controversies regarding its drug safety [36], the metabolites and in vivo hepatotoxicity through bioactivation were intensely studied [58,59,60,61].

As shown in Figure 3, two unusual Noscapine metabolites, hydrocotarnine (M4) and meconine (M5), were observed in the urine samples from animal models (rats and rabbits) and human subjects administered with Noscapine [62]. The formation of these metabolites indicated that one C–C bond in Noscapine is cleaved during its metabolism [63]. In vitro studies using HLM and human P450 isoforms suggested that multiple P450s are involved in this C–C bond cleavage. Among these enzymes, CYP3A4 plays a significant role in Noscapine metabolism [36]. The mechanism (s) of this rare C–C bond cleavage remains unknown, and the mechanism likely involves P450-mediated oxidation to form an unstable intermediate leading to bond cleavage.

### 2.4. Tipranavir

Tipranavir (TPV), a non-peptidic protease inhibitor, is commonly used to treat drug-resistant HIV infections in combination with ritonavir (RTV, CYP3A inhibitor) [37]. Our previous study revealed that an unusual dealkylated metabolite (M6) resulted from C–C bond cleavage in TPV metabolism in mice (Figure 4). Using chemical inhibitors (RTV) in HLM and human P450 isozymes, CYP3A4 was identified as a key enzyme for this C–C bond cleavage reaction [38,64]. M6 was also observed in the incubation of TPV with cDNA recombinant human CYP3A4. The precise mechanism of dealkylation is still unclear, and further research is required to determine the mechanism of the C–C bond cleavage in TPV.

### 2.5. Evatanepag

Evatanepag (also known as CP-533,536), an EP2 receptor-selective prostaglandin E2 agonist, shows promise for aiding bone fracture healing [65]. In HLM, two C-demethylation-related metabolites, M9 and M10, were identified (Figure 5) by LC-MS, and their structures were confirmed by NMR. CYP3A4/5 and CYP2C8 were determined as the major enzymes responsible for the C–C bond cleavage using CYP3A4/5 inhibitors (ketoconazole) and a CYP2C8 inhibitor (quercetin).

The proposed mechanism, as shown in Figure 5, involved the oxidation of the tert-butyl moiety to form alcohol M7, which was further oxidized into aldehyde M8. The resulting M8 produced a CYP3A4/5 mediated carbon-centered radical, which could either undergo oxygen rebound to yield M9 or hydrogen abstraction to create the olefin M10 [39]. Moreover, the possibility of carboxylic acid metabolite as the key intermediate was ruled out because incubating M11 with either HLM or CYP3A/2C8 isoforms did not generate C-demethylated metabolites.

### 2.6. Olanexidine

Olanexidine is an antiseptic containing a biguanide group with a long aliphatic side chain [40,66]. Incubation of Olanexidine in HLM, rat, or dog liver S9 fractions revealed that it could be metabolized into several shortened side chain metabolites (M14 and M15) via C–C bond cleavages (Figure 6) [41].

CYP3A4 is the major enzyme involved in the C–C bond cleavage [40,41].

The proposed mechanism for C–C bond cleavage in Olanexidine involves the formation of a ketol intermediate M12 or M13.

A ferric peroxide (Fe^III^-O-OH) complex is trapped by the electrophilicity of the carbonyl group, leading to a hydroperoxide intermediate. This intermediate subsequently triggers C–C bond cleavage. M12 led to aldehyde M14 by the loss of carboxylic acid, which could be further oxidized to acid M15, while M13 could directly proceed to acid M15 (Figure 6) [41,67]. Incubating ketol intermediates M12 and M13 in HLM and P450s generated the metabolites M14 and M15, supporting the proposed mechanism of C–C bond cleavage in Olanexidine.

### 2.7. Nabumetone

Nabumetone is a widely used non-steroidal anti-inflammatory prodrug [68]. In vitro studies with HLM and P450 isoforms found that CYP3A4 and CYP1A2 are responsible for the formation of acid 6-MNA [42,43,44,69]. The proposed mechanism is that Nabumetone was oxidized to M16, followed by the addition of a nucleophile, the peroxyl anion (Fe^III^-O-O-) intermediate, to the carbonyl group. The resulting tetrahedral intermediate releases carboxylic acid to form an aldehyde M17, which was converted to acid 6-MNA (Figure 7) [44].

A recent study also demonstrated that 6-MNA could be produced via Baeyer–Villiger oxidation catalyzed by flavin-containing monooxygenase isoform 5 (FMO5). Nabumetone was first oxidized to ester (M18) by FMO5, followed by hydrolysis to produce primary alcohol (M19) [45]. The carboxylesterase (CES) inhibiting the formation of 6-MNA in the S9 fractions indicated CES may play a role in the ester hydrolysis as the alcohol is the precursor of 6-MNA formation. The alcohol was oxidized to the intermediate aldehyde M17 and further converted to 6-MNA. As to which pathway plays the dominant role, more studies are needed.

## 3. CYP3A Mediates C(sp^2^)-C(sp^3^) Bond Cleavage

### 3.1. Pexidartinib

Pexidartinib (PEX) is a kinase inhibitor used to treat tenosynovial giant cell tumors, especially when surgery is not a suitable option [70]. However, cases of serious liver toxicity in some patients raised concerns about its safety [71]. In HLM, four metabolites of PEX (M20, M21, M22, and M23) indicated C–C bond cleavages occurred (Figure 8 and Figure 9). The structures of these metabolites were confirmed based on their exact masses, MS/MS fragmentation patterns, and their standard compounds. CYP3A was determined as a primary enzyme responsible for cleavage using human P450 isozymes and chemical inhibitors (ketoconazole, a CYP3A inhibitor) in HLM [46,72].

This study revealed two distinct pathways for C–C bond cleavage in PEX metabolism. The first pathway, as shown in Figure 8, involved CYP3A-activated oxygen added to the 5-alkylated position of a 5-alkylated N-protected pyridin-2-amine to form the intermediate IM1. The substituent elimination of IM1 leads to C–C bond cleavage, yielding phenol M20 and intermediates IM2 or IM3. The formation of M21 and M22 indicated that the intermediates IM2 or IM3 exist. In H_2_^18^O and ^18^O_2_, enriched incubation systems suggested that the source of oxygen in M20 is from the CYP3A-activated oxygen. All the evidence supports that the C–C bond cleavage occurred through a CYP3A-mediated oxygen ipso-addition mechanism.

In the second pathway, as shown in Figure 9, a pseudo-retro-aldol mechanism was proposed. We found that the formation of alcohol M23 was mediated by CYP3A. The beta-hydroxy enamine moiety of M23 (circled) can undergo a pseudo-retro-aldol reaction to produce aldehyde M25 and 5-chloro-7-azaindole (M24). Both metabolites were detected in the incubation system with HLM and CYP3A4 using a PEX substrate. Alcohol M23 as a substrate more efficiently generated M24 and M25, suggesting that M23 is on the kinetic pathway of M25 formation from PEX.

The safety of the metabolites generated from these unusual metabolic pathways is unknown. Thus, this research not only sheds light on the mechanism of novel and rare CYP3A-mediated C–C bond cleavages but also provides valuable insights for drug safety evaluations [46].

### 3.2. Nefazodone

Nefazodone is an atypical antidepressant medication but was withdrawn from some countries due to rare liver toxicity [73]. In HLM, a specific breakdown pathway of Nefazodone involving C–C bond cleavage was identified (Figure 10). Incubating Nefazodone in HLM with CYP3A4 inhibitor and P450s isoenzymes both in vitro [74] and in vivo [75,76] revealed that the metabolite triazoledione formed by a CYP3A4-mediated C(sp^2^)-C(sp^3^) bond cleavage (Figure 10).

The mechanism was proposed as an initial hydroxylation of the ethyl side chain of Nefazodone to form alcohol M26, which undergoes C–C bond scission to generate intermediate alcohol M27. Possibly, M26 was further oxidized to ketone intermediate, which proceeded to M27 by the Baeyer–Villiger reaction. It is well known that M27 is tautomerized to a ketone, triazoledione in the body [47]. The exact mechanism of this C–C bond cleavage is still unclear.

### 3.3. Indacaterol

Indacaterol, a long-acting inhaled β2-adrenergic receptor agonist, is used for the treatment of chronic obstructive pulmonary disease [77]. The metabolite M32 from Indacaterol was identified as a unique metabolic product characterized by an unusual C–C bond cleavage linking the hydroxy-quinolinone and diethyl-indanyl-aminoethanol moieties.

The first potential mechanism pathway involved a one-electron oxidation of the hydroxyquinolinone moiety resulting in the formation of a semiquinone imine radical intermediate M28 (Figure 11). This radical would then migrate to the carbon adjacent to the hydroxyquinolinone group where a hydrogen atom abstraction could occur, followed by a β-scission that leads to the breaking of the C–C bond, forming M30 and M31. The second pathway suggested that an initial hydroxylation at the hydroxyquinolinone M29 happened, followed by a retro-aldol reaction, resulting in the cleavage of the C–C bond. The cleavage formed aldehyde M30, which could then be oxidized to yield the carboxylic acid M32 [48].

CYP3A4 could be the primary enzyme responsible for the formation of metabolite M32 based on the observation that CYP3A4 inhibitor ketoconazole increases serum Indacaterol area under the curve [48].

## 4. CYP3A Mediates Miscellaneous C–C Bond Cleavage

### 4.1. DPC 963

DPC 963 is a non-nucleoside human immunodeficiency virus-1 reverse transcriptase inhibitor (NNRTI) [78]. In rat bile and human liver microsomes, the diastereomeric glutathione adducts M33 were identified (Figure 12), implying that a C–C bond cleavage occurred during the metabolism. The chemical structure of M33 was confirmed by NMR. Using anti-rat P450 antibodies, selective chemical inhibitors, and human P450 isoform enzymes, CYP3A was identified as a major enzyme contributing to the formation of glutathione adducts.

The proposed mechanism of M33 formation involved multiple steps (Figure 12): CYP3A4-mediated oxidation of the triple bond to form the putative oxirene, followed by ring expansion to form a butyl cation via C–C bond cleavage or rearrangement to form the reactive cyclobutenyl ketone [79,80,81]. These active species were converted to an α,β-unsaturated ketone intermediate M34, which reacts with GSH to form the M33 [49]. However, the exact mechanism is still unclear.

The formations of GSH adducts are often considered as detoxification in vivo. Overproduction of the reactive species may cause adverse effects; thus, avoiding co-administration of CYP3A inducers may improve DPC 963 efficacy and safety.

### 4.2. 2,2,6,6-Tetramethylpiperidine

2,2,6,6-Tetramethylpiperidine (TMPi) is a building block with the widespread use of piperidine derivatives in drug development. Researchers discovered an uncommon ring-shrinked metabolite M35, indicating that the TMPi ring undergoes a C–C bond cleavage and ring contraction. NMR spectroscopy experiments confirmed the structure of contracted pyrrolidine ring M35 (Figure 13). Additional experiments with anti-P450 monoclonal antibodies and P450 isoenzymes suggested that the CYP3A4 primarily contributed to this metabolic transformation [51]. The potential mechanism is proposed as a nitroxide radical derived from TMPi interacts with heme iron, leading to homolytic scission of the N-O bond and subsequent C–C bond cleavage, resulting in the contraction of the piperidine ring to M35.

Evidence for Pathway 1 is based on the generation of acetone in the incubation of TMPi with CYP3A4, whereas Pathway 2 provides a possible explanation for the incorporation of two oxygen atoms into the molecule from the water of the incubation medium using a different H_2_^18^O ratio before its final degradation to 2,2-dimethylpyrrolidine. This transformation could occur through CYP3A4 catalysis or other heme proteins, suggesting a heme-catalyzed process [51]. Thus, in the study of the metabolism of TMPi-containing drugs, these uncommon metabolites should be monitored.

### 4.3. Cipargamin

Cipargamin (also known as KAE-609) is a drug candidate under clinical development by Novartis and is currently in Phase II for Malaria [82]. In the metabolic study of Cipargamin in healthy male subjects [54], rats, and dogs, a notable metabolite M36 (Figure 14) was found. In the proposed mechanism, the C–C bond cleavage-mediated ring expansion process is involved in the formation of M36 [55]. Briefly, an initial oxidation step by CYP3A4 results in the formation of a radical cation. Following the ring opening via C–C bond cleavage, subsequent recyclization produced metabolite M36 (Figure 14) [55]. CYP3A4 is mainly responsible for C–C bond cleavage in the formation of M36. The exact mechanism of this C–C bond cleavage is still unknown.

## 5. Summary

In the pharmaceutical field, identifying the uncommon reactions in drug metabolism, like C–C bond cleavages, is challenging but essential for understanding drug interactions and improving drug safety. In this review, we especially summarize the drugs in which CYP3A-mediated C–C bond cleavage occurred, including C(sp3)-C(sp3), C(sp2)-C(sp3), and other uncommon reactions, along with the available mechanisms. Determining the mechanisms of the C–C bond cleavages could provide hints for predicting this type of uncommon reaction. The insights gained from studying CYP3A-mediated carbon–carbon (C–C) bond cleavage have significant implications for drug design, safety, and pharmacokinetics. Overall, a thorough investigation of drug metabolism will ensure the safe and effective use of pharmaceuticals, including unusual reactions like CYP3A-mediated C–C bond cleavages. Additionally, investigating the detailed mechanisms underlying C–C bond cleavage will further expand our understanding of CYP3A functions.

## Figures and Tables

**Figure 1 biomolecules-14-01125-f001:**
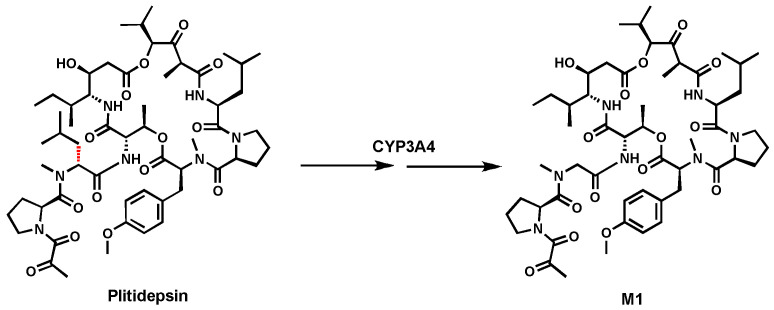
CYP3A4 mediates C(sp^3^)-C(sp^3^) bond cleavage in Plitidepsin.

**Figure 2 biomolecules-14-01125-f002:**
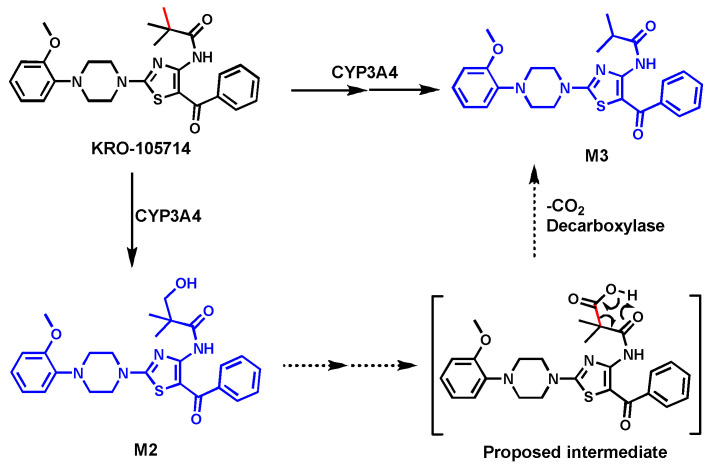
C-Demethylation of KRO-105714.

**Figure 3 biomolecules-14-01125-f003:**
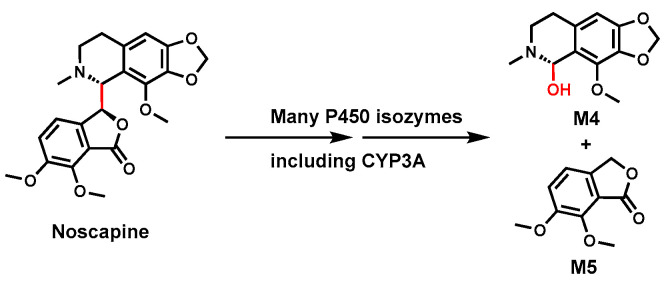
CYP3A4 mediates C(sp3)-C(sp3) bond cleavage in Noscapine.

**Figure 4 biomolecules-14-01125-f004:**
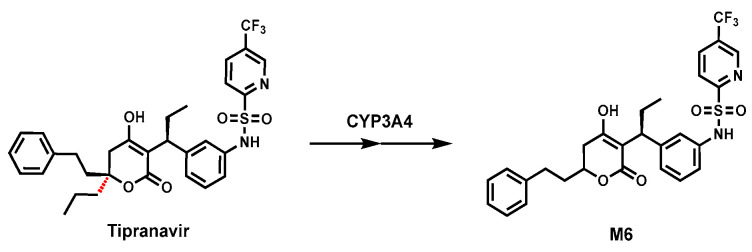
C-Depropylation of Tipranavir.

**Figure 5 biomolecules-14-01125-f005:**
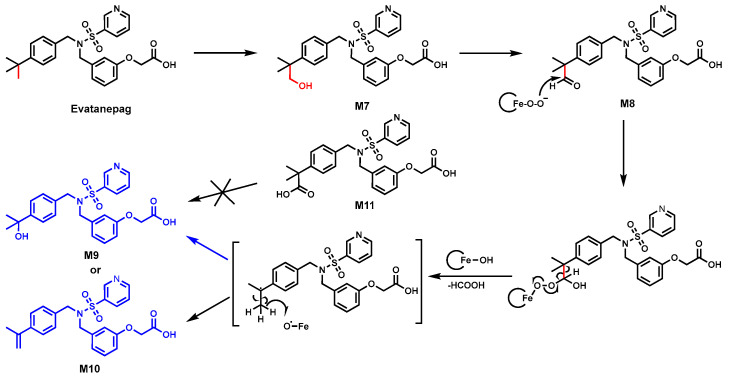
Proposed mechanism for demethylation of the t-butyl group in Evatanepag.

**Figure 6 biomolecules-14-01125-f006:**
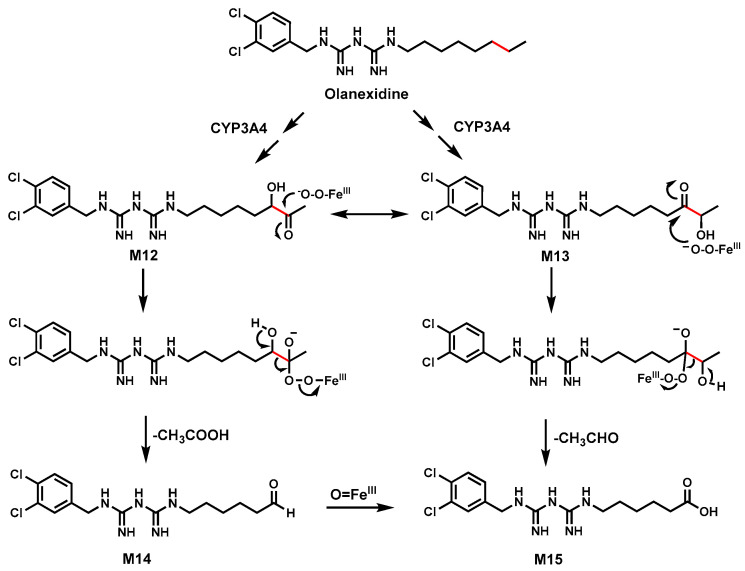
The proposed mechanism for carbon–carbon bond cleavage in Olanexidine.

**Figure 7 biomolecules-14-01125-f007:**
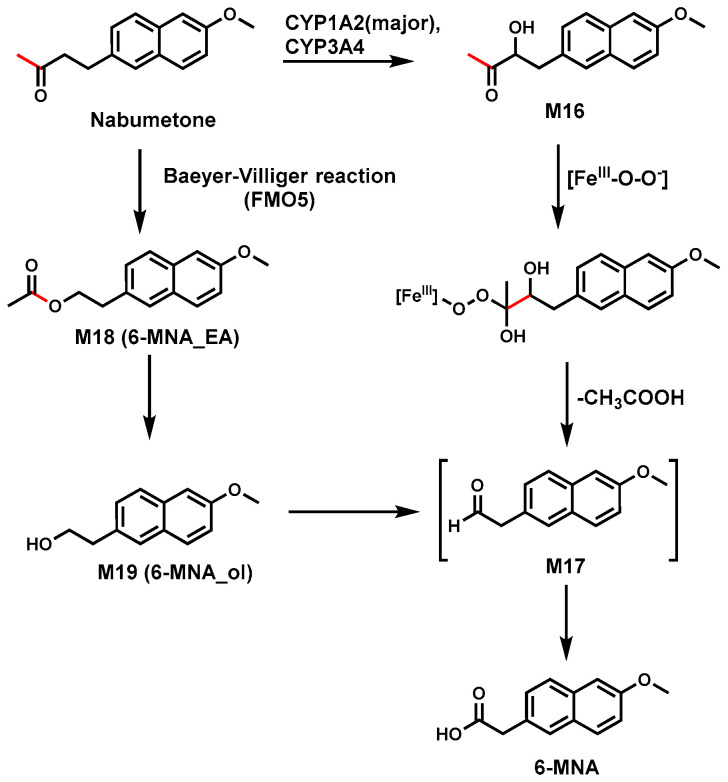
Proposed mechanism for the C–C bond cleavage in Nabumetone.

**Figure 8 biomolecules-14-01125-f008:**
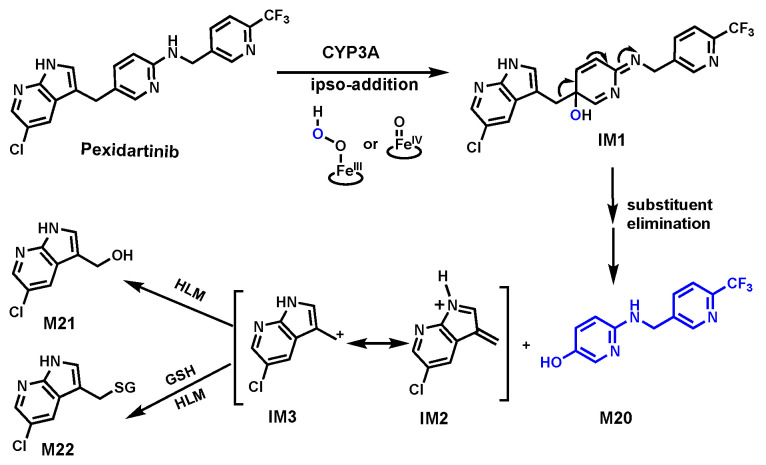
CYP3A-mediated C–C bond cleavage in Pexidartinib via ipso-addition of oxygen.

**Figure 9 biomolecules-14-01125-f009:**
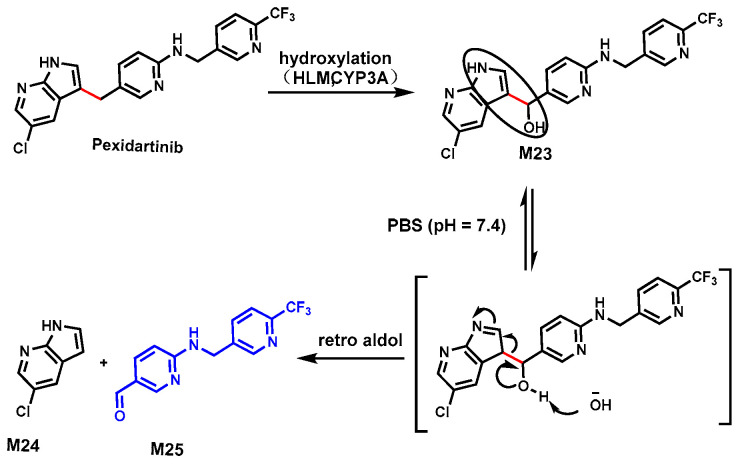
CYP3A-mediated C–C bond cleavages in Pexidartinib via retro aldol.

**Figure 10 biomolecules-14-01125-f010:**
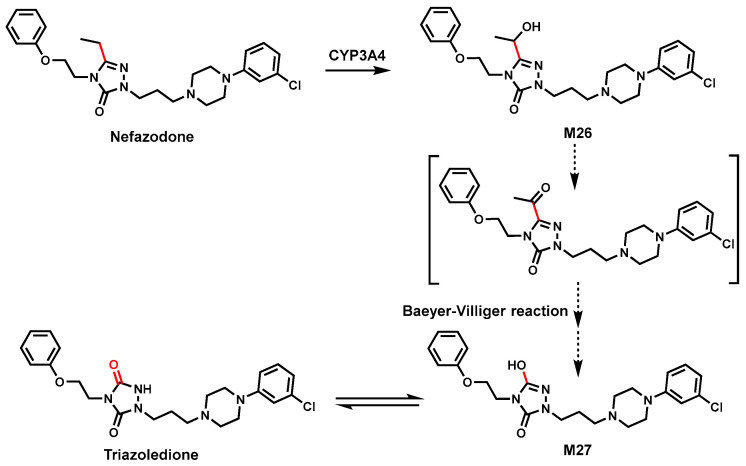
CYP3A4 mediates C–C bond cleavage in Nefazodone.

**Figure 11 biomolecules-14-01125-f011:**
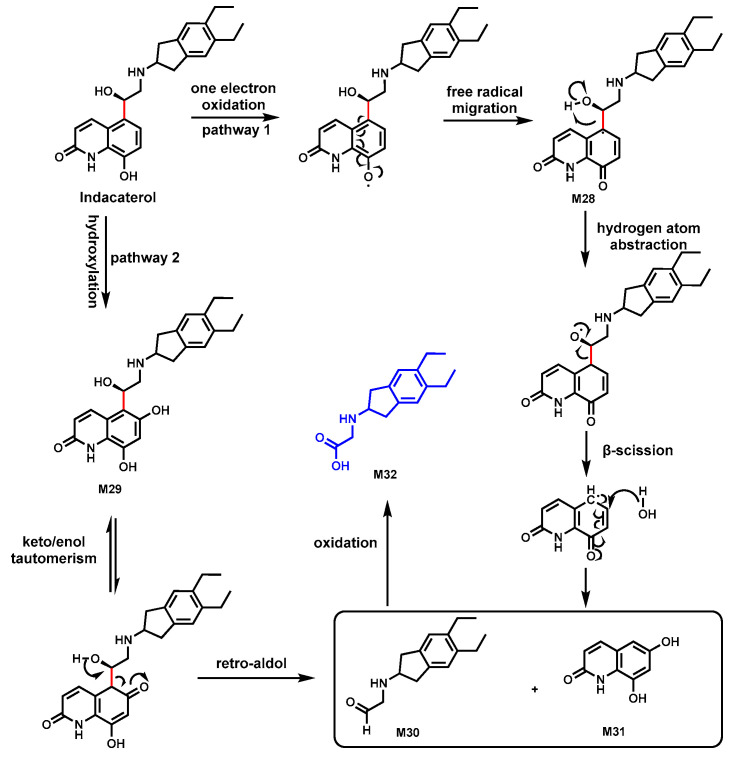
Proposed mechanism of M30 formation from Indacaterol.

**Figure 12 biomolecules-14-01125-f012:**
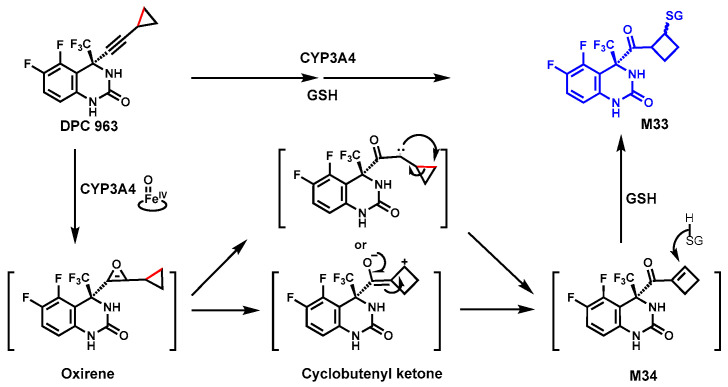
CYP3A4-mediated ring expansion of the cyclopropyl ring in DPC 963.

**Figure 13 biomolecules-14-01125-f013:**
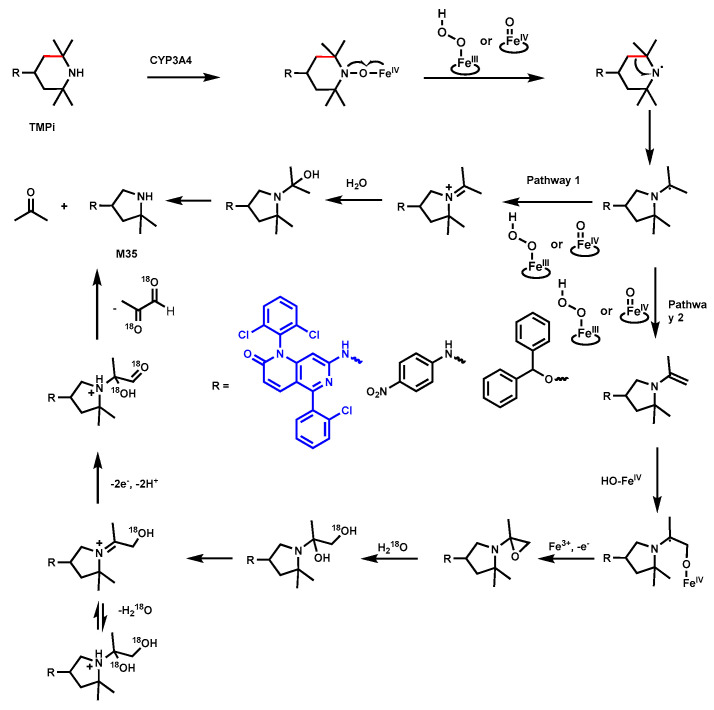
Carbon–carbon bond cleavage in Tetramethyl-piperidine.

**Figure 14 biomolecules-14-01125-f014:**
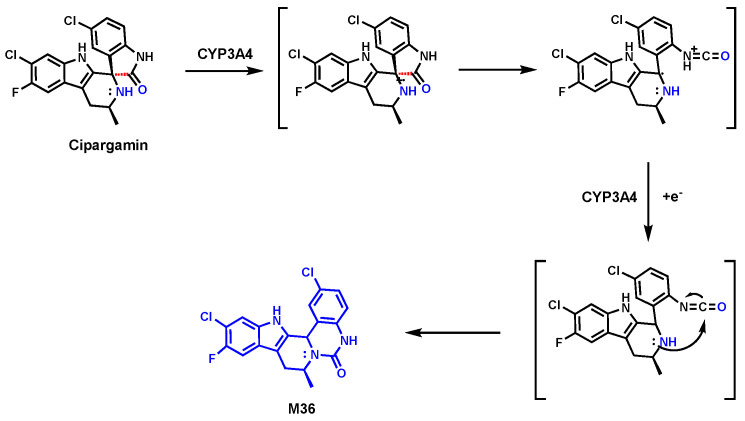
C–C bond cleavage in Cipargamin.

**Table 1 biomolecules-14-01125-t001:** Drugs and the types of C–C bond cleavage.

Drug Name	Indications	C–C Bond Cleavage Type
Plitidepsin	Antitumor and antiviral activities	C(sp^3^)-C(sp^3^) [30,31,32,33]
IKRO-105714	Alleviate atopic dermatitis	C(sp^3^)-C(sp^3^) [34,35]
Noscapine	Cough suppressing	C(sp^3^)-C(sp^3^) [36]
Tipranavir	Anti-HIV agent	C(sp^3^)-C(sp^3^) [37,38]
Evatanepag	Treats tibial fractures.	C(sp^3^)-C(sp^3^) [39]
Olanexidine	Antiseptic agent	C(sp^3^)-C(sp^3^) [40,41]
Nabumetone	Treats pain and arthritis	C(sp^3^)-C(sp^3^) [42,43,44,45]
Pexidartinib	Treats tenosynovial giant cell tumor	C(sp^2^)-C(sp^3^) [46]
Nefazodone	Treats depression	C(sp^2^)-C(sp^3^) [47]
Indacaterol	Treats chronic obstructive pulmonary disease and asthma	C(sp^2^)-C(sp^3^) [48]
DPC 963	Anti-HIV agent	Ring-open [49,50]
Tetramethylpiperidine	Building blocks of drugs	Ring-open [51,52,53]
Cipargamin	Anti-Malarial	Ring-open [54,55]

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
