# Peer review of "CYP3A-Mediated Carbon–Carbon Bond Cleavages in Drug Metabolism"

_biomolecules, 2024, doi:10.3390/biom14091125_

Round 1
Reviewer 1 Report
Comments and Suggestions for Authors
The manuscript of Mackenzie and coll. present an interesting review on reported C-C cleavage catalyzed by CYP3A enzymes.
They make a good compilation of published work and try to discuss the possible mechanisms involved in these metabolic reactions. They discuss 13 examples.
The paper is well written and easy to read.
I have a point on the format of the references. I think for a review you should put all the name of the authors. Otherwise it is difficult to find studies done by a group since often the leader is the last author and thus disappears in the used format.
Concerning the figures, I think it would be nice to show which molecules have been really isolated. For instance you could use colour.
Along the review I made some comments as below:
For KRO-105714 Figure 2 : You form a maloamide a type of structure that is easily decarboxylated chemically in acid medium.
There may be a catalyst for that reaction in cells.
Figure 5. Proposed mechanism for demethylation of the t-butyl group in Evatanepag : One could use the mechanism of deformylation of aromatase of CYP51. See numerous recent studies in Guengerich group (that you cite).
Probably the acid does not have a good affinity for CYP3A4. The aldehyde being neutral may have a good affinity. Then the mechanism first described by Coon may apply. doi: 10.1073/pnas.88.20.8963 , doi: 10.1073/pnas.93.10.4644.
Figure 6 : are the compounds M12 M13 isolated and characterized? Otherwise what about simple omega oxidation to the acid and then beta-decarboxylation?
Figure 7 : correct Baeyer-Villiger
Figure 11 : I don't understand the moving of the arrows in formation of the retroaldol?
Figure 12 presents a puzzling study of Mutlib and colleagues. Thus the formatting of the references is disappointing since you cannot find a number of references by his name.
In Figure 13. Carbon-carbon bond cleavage in Tetramethyl-piperidine: you must show which compounds have been confirmed in this scheme. (perhaps using color)
Alltogether the paper is original and quite interesting and should be published after minor corrections. (I really think the formatting of the references should be changed)
Reviewer 2 Report
Comments and Suggestions for Authors
see attached

Comments on the Quality of English Languageneeds native English speaker editing.
Round 2
Reviewer 2 Report
Comments and Suggestions for Authors
The manuscript looks better than the original submission. Here are some points to help authors improve the manuscript and I think it should be fine after the corrections:
Page 10 of 17 - 2,2,6,6-Tetramethylpiperidine in the paragraph beginning of the sentence, the T should be capitalized.
Figure 6- going from M13 to M15 - the molecule that is being released should be acetaldehyde not acetic acid as the authors have shown.
Figure 9 - M23 to the structure in brackets - the arrow should be equilibrium arrows instead of a "resonance structure arrow" since the proton is also being shuffled (resonance structure arrows should be used only if the electrons are moving around in the molecule).
Figure 10 - M27 to triazoledione structure should have equilibrium arrows instead of the resonance structure arrow since protons are moving around in addition to electrons.
Comments on the Quality of English Language
Page 10 of 17 - line 246 - sentence starts with "GSH formation are" - the verb used is plural but the subject is singular (GSH formation) authors need to change the subject to plural as "GSH formations"
Page 4 of 17 - line 114-115, should be "and a CYP2C8 inhibitor (quercetin)" (missing the word "a" before the noun).
Page 4 of 17 - line 104-105: should be "for this C-C bond cleavage reaction." Also in this paragraph - authors are mentioning the study done in mice - was this done with the mouse CYP3A4? Then did the authors perform the same incubation with the human isoenzyme - did they separately do the reaction with human CYP3A4 and find the same product? It should probably be clarified.
Line 71, Page 2 of 17 - a specific inhibitor of CYP3A4 was mentioned but not stated - what was the specific CYP3A4 inhibitor used? Was it ketoconazole? If so - this is a broad P450 inhibitor so the concentration is important if it was being selective for CYP3A4. What was the concentration of the specific inhibitor used here?
